# Universal bounds on transport in holographic systems with broken translations

**Matteo Baggioli[1*] and Wei-Jia Li[2]**

**1** Instituto de Fisica Teorica UAM/CSIC, c/Nicolas Cabrera 13-15,
Universidad Autonoma de Madrid, Cantoblanco, 28049 Madrid, Spain.
**2** Institute of Theoretical Physics, School of Physics,
Dalian University of Technology, Dalian 116024, China.

* matteo.baggioli@uam.es

## Abstract

We study the presence of universal bounds on transport in homogeneous holographic models with broken translations. We verify numerically that, in holographic systems with momentum dissipation, the viscosity to entropy bound might be violated but the shear diffusion constant remains bounded by below. This confirms the idea that $\eta/s$ loses its privileged role in non-relativistic systems and that, in order to find more universal bounds, one should rather look at diffusion constants. We strengthen this idea by showing that, in presence of spontaneously broken translations, the Goldstone diffusion constant satisfies a universal lower bound in terms of the Planckian relaxation time and the butterfly velocity. Additionally, all the diffusive processes in the model satisfy an upper bound, imposed by causality, which is given in terms of the thermalization time – the imaginary part of the first non-hydrodynamic mode in the spectrum – and the speed of longitudinal sound. Finally, we discuss the existence of a bound on the speed of sound in holographic conformal solids and we show that the conformal value acts as a lower (and not upper) bound on the speed of longitudinal phonons. Nevertheless, we show that the stiffness $\partial p/\partial \epsilon$ is still bounded by above by its conformal value. This suggests that the bounds conjectured in the past have to be considered on the stiffness of the system, related to its equation of state, and not on the propagation speed of sound.

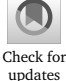

# 1   Introduction

> Living is dangerous. The important thing is to know the limits.
>
> Andrea Bocelli

*How fast can a wave propagate?   How chaotic a quantum mechanical system can become? How runny a liquid can be?*  Imposing universal bounds on physical processes is one of the Holy Grail of human curiosity and one of the more noble task of science.

Several of these bounds originate from the most fundamental theories of physics – special relativity and quantum mechanics – and they are connected to important concepts such as causality or the Heisenberg uncertainty principle [1]. The latter, in particular, leads to the definition of the so-called Planckian time [2]:

$$\tau_{pl} \equiv \frac{\hbar}{k_B\, T}\,, \tag{1}$$

which sets the minimal time-scale available in nature [3], and it plays a key role in several quantum mechanical bounds and physical processes [4–17].

The notion of a minimal relaxation time appeared first in the definition of the famous viscosity to entropy Kovtun-Son-Starinets (KSS) bound [8]:

$$\frac{\eta}{s} \geq \frac{1}{4\,\pi}\frac{\hbar}{k_B}\,, \tag{2}$$

which is so far respected by all the systems we know in nature [18–22]. The value of this ratio is taken as a valid indicator of how strongly coupled a system is. The smallest value in nature is found in the quark gluon plasma and it is approximately three times larger than the KSS value [18]. A similar bound on the kinematic viscosity of liquids has been formulated in terms of fundamental constants [23] and it has revealed surprisingly similarities between common liquids and the quark gluon plasma [24].

The KSS bound (2) was later generalized by Hartnoll [25] as a bound on transport diffusion constants:

$$D \geq v^2 \tau,$$ (3)

where $v$ is a characteristic velocity of the system and $\tau$ the relaxation time. Notice how the KSS bound can be embedded in this new formulation using the equivalence:

$$\frac{\eta}{s} \frac{k_B}{\hbar} = \frac{D_T}{\tau_{pl}},$$ (4)

where the characteristic speed is simply the speed of light $c$, which is set for simplicity to unit, and $D_T$ refers to the momentum diffusivity.

In a sense, this second formulation (3), is much more general than (2) because it applies to generic diffusion processes (not necessarily related to the momentum dynamics – viscosity) and because it provides a robust definition also in non-relativistic systems. Nevertheless, at first sight, it seems quite void because it does not provide any specifics about which is the velocity scale involved. One of the strongest motivations of [25] was to rationalize the linear in $T$ resistivity of strange metals [26], where indeed the role of the universal Planckian relaxation time has been experimentally observed [4]. In those systems, the quasiparticle nature is lost because of strong interactions/correlations and, as an immediate consequence, the Ioffe-Regel limit is brutally violated at high temperatures [27]. Given this observation, the velocity appearing in Eq.(3) cannot be generically identified as the microscopic velocity of the quasiparticles, e.g. the Fermi velocity for a weakly coupled electron fluid.

Borrowing from quantum information concepts, Blake [28,29] proposed to identify that velocity scale with the *butterfly velocity*, defined via the OTOC – out of time ordered correlator [30]. In simple words, the butterfly velocity $v_B$ determines the information scrambling speed in a quantum mechanical system and it can be defined robustly even in absence of quasiparticles. The refined bound:

$$D \geq v_B^2 \tau_{pl}$$ (5)

has been tested and discussed in a large number of works especially in connection to black hole physics, holography, quantum information and SYK physics [14,31–49].

The final outcome is that, if one considers charge diffusion, the bound in (5) can be violated by considering the effects of inhomogeneity [50] or higher derivative corrections [51], in a way similar to the violation of the conductivity bound of [52], which was shown in [53] and later in [54]. Nonetheless, if the bound (5) is applied to energy diffusion:

$$D_e \equiv \frac{\kappa}{c_v},$$ (6)

with $\kappa$ the thermal conductivity and $c_v$ the specific heat, only one subtle violation has been found so far in the context of SYK chains [55]. On the contrary, all holographic models obey the bound on energy diffusion. Along the same lines, universality was also discovered in the Chern-Simons diffusion rate, in a large class of planar strongly correlated gauge theories with dual string description [56] and in the Langevin diffusion coefficients [57].

From another perspective, we can ask ourselves "how fast" a diffusive process can occur in nature. The authors of [15] found that causality, intended as the absence of superluminal motion, bounds from above any diffusion constant:

$$D \leq v_{\text{lightcone}}^2 \tau_{eq},$$ (7)

where $v_{\text{lightcone}}$ is the lightcone velocity, which defines the concept of causality, and $\tau_{eq}$ the equilibration or thermalization time at which diffusion sets in. Moreover, Ref. [15] showed

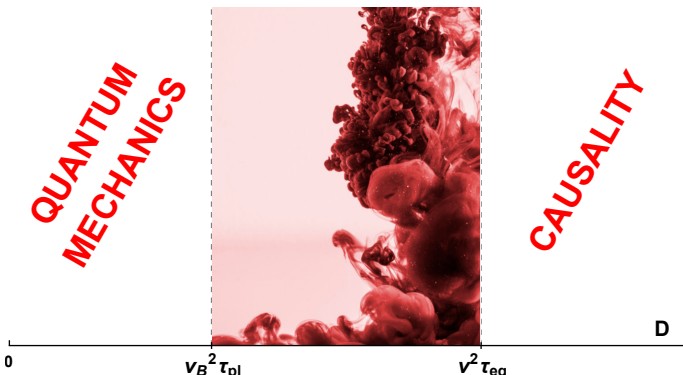

Figure 1: The lower and upper bounds on the diffusion constant $D$. Here, $v_B$ stands for the butterfly velocity, $v$ is the lightcone speed, $\tau_{pl}$ the Planckian time and $\tau_{eq}$ the equilibration time.

that this bound is obeyed in simple holographic systems with Einstein gravity duals.

All in all, the discussions above suggest that the diffusion constant of a generic diffusive process is bounded both from below and from above, but by different timescales and because of different physical reasons. We summarize the generic expectations for diffusion in Fig.1.

Despite the incredible success of the KSS bound on the viscosity to entropy ratio, things become more complicated when spacetime symmetries are broken and the holographic models are made more realistic. It was already observed long time ago [58] that the explicit (but, curiously, not the spontaneous [59]) breaking of rotations produces a violation of the KSS bound (2). More recently, the violation of the KSS bound was observed in some holographic massive gravity models [60–62], where momentum is not conserved anymore. Here, we want to address the following questions:

- What is the meaning of this violation?

- Is this violation due to momentum dissipation? And, more in general, how is that connected to the breaking of spacetime symmetries?

- What happens if we consider the more generic bound on the momentum diffusivity (3), instead of considering the $\eta/s$ ratio?

- Is the violation appearing also if translations are spontaneously broken?

- Are there other quantities which can universally bounded in holographic systems with spontaneously broken translations?

Spoiling the results of the next sections, we conclude that: **(I)** the momentum diffusion constant is bounded by below also in systems with momentum dissipation; **(II)** the $\eta/s$ bound, differently from the rotational symmetry case, is violated independently of the explicit or spontaneous breaking of translational invariance; **(III)** the crystal diffusion constant obeys a universal bound as in Eq.(5); **(IV)** a new way to violate the universal bound for charge diffusion is by introducing phonons dynamics, and coupling the charge sector to the Goldstone one. In this case, the diffusion constant which reduces to charge diffusion at zero coupling violates the standard bound. **(V)** All the diffusive processes in the holographic models with spontaneously broken translations obey an upper bound on diffusion, where the lightcone speed is the speed of longitudinal sound and the thermalization time is extracted as the imaginary part of the first non-hydrodynamic mode.

Until this point, we have focused our discussion on the diffusive dynamics, which is typical of incoherent systems (e.g. systems where momentum is dissipated very fast) or systems with conserved quantities (e.g. charge conservation, energy conservation, etc). Nevertheless, specially in systems with long-range order, there is an additional low-energy dynamics which is governed by the so-called sound modes, linearly propagating modes:

$$\omega = v_s k - i \frac{\Gamma_s}{2} k^2 + \dots,$$ (8)

where $v_s$ is the speed of sound and $\Gamma_s$ the sound attenuation constant. The "..." emphasize that this dispersion relation is valid only at low momenta and it receives corrections at short length-scales.

This happens also in standard quantum fluids [63], where the speed of (longitudinal) sound is given by[1]:

$$v_s^2 = \frac{\partial p}{\partial \epsilon},$$ (9)

with $p$ and $\epsilon$ respectively the pressure and the energy density. In such context, Refs. [65, 66] conjectured a universal upper bound on the speed of sound, where the upper limit is given by the conformal value:

$$v_c^2 = \frac{1}{d-1},$$ (10)

with $d$ the number of spacetime dimensions.

More recently, Refs. [10, 67] proposed different upper bounds on the speed of sound in solid systems with long-range order, where the sound speed is determined by the elastic moduli of the material [68].

In the last years, several holographic models exhibiting elastic properties and propagating phonons have been introduced [69–73]. In this work, we ask the very simple question: is the speed of sound bounded in these models and by what? Is the conformal value really the upper limit? We will discuss in detail the numerical results and the implications given by the absence of a UV cutoff, playing the role of the lattice spacing or the Debye temperature.

In all our discussions, we will only consider holographic models in the large $N$, strong coupling, limit in which the gravitational dual is described by weakly-coupled classical gravity. It is well known (see [19] for a review) that $1/N$ corrections can induce a violation of the KSS bound [8] and push the ratio $\eta/s$ below its "universal" value of $1/4\pi$. Nevertheless, the $1/N$ corrections are performed only in a perturbative way and they are subject to strong constraints coming from causality and other consistency conditions. As a result of that, the violations of the KSS bound induced by these effects are not parametric (like in the case of broken spacetime symmetries which we consider in this work) but they just modify the $\mathcal{O}(1)$ constant # appearing in $\eta/s \geq \#\hbar/k_B$, which does not have any fundamental meaning.

## 2 The class of holographic models

We consider a large class of holographic models with broken translations which was introduced in [74, 75] and defined by the following 4-dimensional bulk action:

$$S = \int d^4 x \sqrt{-g} \left[ \frac{R}{2} + 3 - m^2 V(X) - \frac{1}{4} F^2 \right],$$ (11)

---

[1]This formula is valid only for uncharged relativistic fluids [64].

where $X \equiv \frac{1}{2} g^{\mu\nu} \partial_\mu \phi^I \partial_\nu \phi^I$ and $F^2 \equiv F_{\mu\nu} F^{\mu\nu}$ together with the usual definition for the EM field strength $F = dA$. For simplicity, we have put both the Planck mass and the AdS radius to be unit. Detailed discussions about this class of models can be found in [76, 77]. The bulk profile for the scalars, which breaks translational invariance in both the spatial directions by retaining the rotational group $SO(2)$, is

$$\phi^I = x^I, \tag{12}$$

and it represents a trivial solution of the equations of motion thanks to the global shift symmetry $\phi^I \to \phi^I + b^I$ of the action (11). The gauge field profile is simply

$$A_t = \mu - \rho u, \tag{13}$$

where $\mu, \rho$ are respectively the chemical potential and the charge density of the dual field theory. We assume an asymptotically AdS bulk line element:

$$ds^2 = \frac{1}{u^2} \left[ -f(u) dt^2 - 2 dt du + dx^2 + dy^2 \right], \tag{14}$$

with $u \in [0, u_h]$ the radial holographic direction spanning from the boundary $u = 0$ to the horizon, $f(u_h) = 0$. The emblackening function is given by:

$$f(u) = u^3 \int_u^{u_h} dv \left[ \frac{3}{v^4} - \frac{V(v^2)}{v^4} - \frac{\rho^2}{2} \right]. \tag{15}$$

The associated dual field theory temperature reads:

$$T = -\frac{f'(u_h)}{4\pi} = \frac{6 - 2V\left(u_h^2\right) - \rho^2 u_h^4}{8\pi u_h}, \tag{16}$$

while the entropy density is given by $s = 2\pi/u_h^2$.

In the rest of the paper, we will focus on a sub-class of models given by the monomial form:

$$V(X) = X^N. \tag{17}$$

The quasinormal modes and the transport coefficients of the system can be obtained by solving numerically the equations for the perturbations. We refer to the previous literature [69, 78–84] for all the details pertaining such computations in the class of models considered in this work. The role of the parameter $N$ (not the rank of the gauge group) is fundamental in determining the nature of the translational invariance breaking. This can be seen by looking at the UV expansion of the scalar bulk fields $\phi^I$ whose profile (constant in the radial coordinate) breaks the translational symmetry. More precisely, the expansion reads as:

$$\phi^I(t, x, u) = \phi^I_{(0)}(t, x) + \phi^I_{(1)}(t, x) u^{5-2N} + \dots \tag{18}$$

Assuming standard quantization, it is straightforward to realize that for $N < 5/2$ the bulk profile $\phi^I = x^I$ represents an external source for the field theory, while for $N > 5/2$ it describes a finite expectation value for the scalar operators $\mathcal{O}^I$ dual to the bulk fields $\phi^I$ in the absence of a source. For this simple reason, the breaking of translations is explicit for $N < 5/2$ and spontaneous for $N > 5/2$. Notice that in the case of a polynomial potential, the smallest power is what determines the UV asymptotics and therefore the nature of the symmetry breaking pattern.

# 3 A lower bound on diffusion

## 3.1 Shear diffusion with momentum dissipation

We start by considering our benchmark model (17) with $N < 5/2$. This corresponds to having a non-trivial and space-dependent source $\Phi_0^I = x^I$ for the scalar operators dual to the bulk fields $\phi^I$. Consequently, this amounts to introduce an explicit breaking of translations invariance and a relaxation time for the momentum density operator [85–87].

Surprisingly, Refs. [60] and [61] simultaneously discovered the violation of the KSS bound [8] in these systems, which was also investigated later in [62,88]. Importantly, the physical reason behind this result and its validity remains not understood. Few violations were observed in field theory as well [89,90].

Several comments are in order.

(I) The definition of viscosity itself in standard hydrodynamics [91] comes from the conservation of the stress tensor, $\nabla_\mu T^{\mu\nu} = 0$. Therefore, its meaning remains robust only in the regime where the momentum relaxation time is long enough, i.e. $\tau T \gg 1$. Nevertheless, perturbative analytic computations [60–62] showed that, even in that regime, the KSS bound is clearly violated.

(II) From a more general perspective, valid even in absence of a well-defined and long-living momentum, the $\eta/s$ ratio can be physically understood as the logarithmic entropy production rate produced by a linear in time deformation [61]:

$$\frac{\eta}{s} \sim \frac{1}{T} \frac{d \log s}{dt}. \tag{19}$$

Bounds on entropy production have been widely discussed in the literature [92–95]. It would be interesting to understand them better under the light of Holography.

(III) In presence of momentum relaxation, the shear diffusion constant is not anymore uniquely determined by the $\eta/s$ ratio. Nonetheless, notice this is also the case for a charged fluid with conserved momentum [64]:

$$D_T = \frac{\eta}{s\,T + \mu\rho}, \tag{20}$$

for which the KSS bound is perfectly respected.

(IV) In presence of broken symmetries, it is not clear anymore that the shear viscosity $\eta$ can be still defined in the same way using the standard Kubo formula in terms of the stress tensor:

$$\eta \overset{?}{=} -\lim_{\omega \to 0} \frac{1}{\omega} \text{Im} \left[ \mathcal{G}^{(\text{R})}_{T_{xy} T_{xy}}(\omega) \right], \tag{21}$$

as pointed out in [62]. Notice that this ambiguity is not there for the case of the SSB of translations and that Ref. [62] found a violation of the KSS bound independent of the definition of $\eta$ (see therein for details). Therefore, we can exclude the possibility that the violation is simply due to a wrong definition of the transport coefficient.

(V) The violation of the KSS bound cannot be simply attributed to the dissipation of momentum since Ref. [60] found a large class of holographic models where momentum is not conserved but the KSS bound holds as usual. This was later confirmed numerically

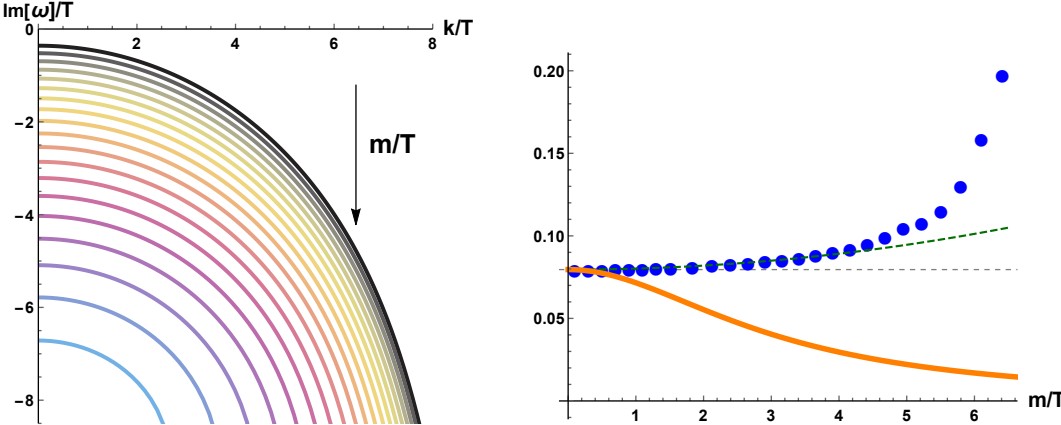

Figure 2: The dynamics of the shear mode in the simple "*linear axions model*" corresponding to $V(X) = X$. **Left:** The dispersion relation of the pseudo-diffusive mode $\omega = -i\Gamma - iD_T k^2$ for $m/T \in [1, 6.5]$ (from black to light blue). **Right:** In orange the viscosity to entropy ratio $\eta/s$, in blue the dimensionless shear diffusion constant $D_T T$ obtained numerically and in green the analytic formula (28). The horizontal dashed value is $1/4\pi$.

in [83]. The crucial point is that these holographic theories with broken translations are not Lorentz invariant and the various components of the graviton might get different masses. In particular the shear mode mass $m_{xy}^2$ can differ from the $m_{tx}^2$ mass which is responsible for momentum dissipation. This is exactly what happens in the so-called fluid theories [60,75], where $m_{xy}^2 = 0$ (which implies the saturation of the KSS bound) but $m_{tx}^2 \neq 0$ (which implies the finiteness of the DC conductivities and the dissipation of momentum). In other words, the key is simply the mass for the shear component of the graviton, as showed generically in [61].

(VI) The original definition of the KSS bound, Eq.(2), works extremely well for relativistic systems but it becomes quite unnatural when relativistic symmetries are abandoned. For non-relativistic fluids, it is well-known that momentum diffusion is controlled by $\eta/\mathfrak{r}$, with $\mathfrak{r}$ being the mass density of the system [91], and not by the $\eta/s$ ratio. This is consistent with the idea proposed by Hartnoll [25] that a better quantity to bound is the momentum diffusion constant and not the $\eta/s$ ratio. Contrarily to the latter, the first observable is universally well-defined, independently of the symmetries of the system. On the same line of thoughts, Ref. [24] found that the kinematic viscosity $\nu$ (which, we remind, corresponds to the momentum diffusivity) of quark gluon plasma at its minimum is not far from the universal value for standard liquids:

$$\nu_m = 10^{-7} \text{ m}^2/\text{s}, \tag{22}$$

making Hartnoll's proposal more quantitative.

Taking into account all these points, here, we study the dynamics of the $\eta/s$ ratio and the momentum diffusion constant $D_T$ in holographic systems with explicitly broken translations. Because of the non-conservation of momentum, the shear diffusive mode acquires a finite damping, $\Gamma = \tau^{-1}$, and its dispersion relation becomes:

$$\omega = -i\,\Gamma - i\,D_T\,k^2 + \mathcal{O}(k^3), \tag{23}$$

as shown in the left panel of Fig.2.

From previous studies [60–62], it is known that if we define the shear viscosity as:

$$\eta \equiv - \lim_{\omega \to 0} \frac{1}{\omega} \operatorname{Im} \left[ \mathcal{G}^{(R)}_{T_{xy} T_{xy}}(\omega) \right], \tag{24}$$

the KSS bound is brutally violated. For a potential $V(X) = X^N$, we can obtain a perturbative formula [60], valid for small graviton mass, which reads:

$$\frac{\eta}{s} = \frac{1}{4\pi} \left( 1 - \frac{4}{3} m^2 u_h^{2N} \frac{\mathcal{H}_{\frac{2N}{3}-1}}{2N-3} \right) + \mathcal{O}(m^4), \tag{25}$$

where $\mathcal{H}_i$ is the $i^{th}$ Harmonic number. Moreover, at small temperature, the ratio displays a scaling of the type:

$$\frac{\eta}{s} \sim \left( \frac{T}{m} \right)^2 \quad for \quad T \to 0, \tag{26}$$

which solely depends on the conformal dimension of the $T_{xy}$ operator in the IR scale invariant geometry [88].

On the contrary, if we consider the momentum diffusion constant appearing in the dispersion relation (23), we can still find a universal bound

$$D_T T \geq \frac{1}{4\pi}, \tag{27}$$

in agreement with Hartnoll's observations [25].
At small graviton mass, we can obtain the following approximate formula [96]

$$D_T T = \frac{1}{4\pi} \left( 1 + \frac{1}{24} \left( 9 + \sqrt{3}\pi - 9\log 3 \right) \frac{m^2}{8\pi^2 T^2} \right), \tag{28}$$

where, different from Eq.(25), the corrections steaming from a finite graviton mass are positive! This result appeared already in [96], but there the authors erroneously identified the shear viscosity as:

$$\frac{\eta}{s} \equiv D_T T, \tag{29}$$

which is certainly not correct. Nevertheless, the idea that the momentum diffusion constant is bounded by below even in presence of momentum dissipation was already hidden in the computations of [96] in the limit of slow dissipation, i.e. $m/T \ll 1$.

Here, we go beyond their approximate solution and we compute the momentum diffusion constant numerically at all orders in the graviton mass. The results are shown in Fig.2. We can observe that the $\eta/s$ ratio drops below the KSS value of $1/4\pi$ by increasing the momentum dissipation strength $\sim m/T$ and it eventually drops to zero at $T \to 0$ in a power-law fashion. On the contrary, the momentum diffusion constant increases with $m/T$ from its initial value $D_T = 1/4\pi T$. At small $m/T$, the numerical data are well approximated by the perturbative formula (28) derived in [96].

Our results confirm the idea that the $\eta/s$ ratio does not hold a privileged role in presence of momentum dissipation since it does not determine any transport coefficients. Moreover, as expected from general arguments [25], the momentum diffusion constant is bounded by below even in the presence of momentum dissipation. This indicates that the violation of the KSS bound found in [60, 61] does not have any fundamental physical meaning. The same

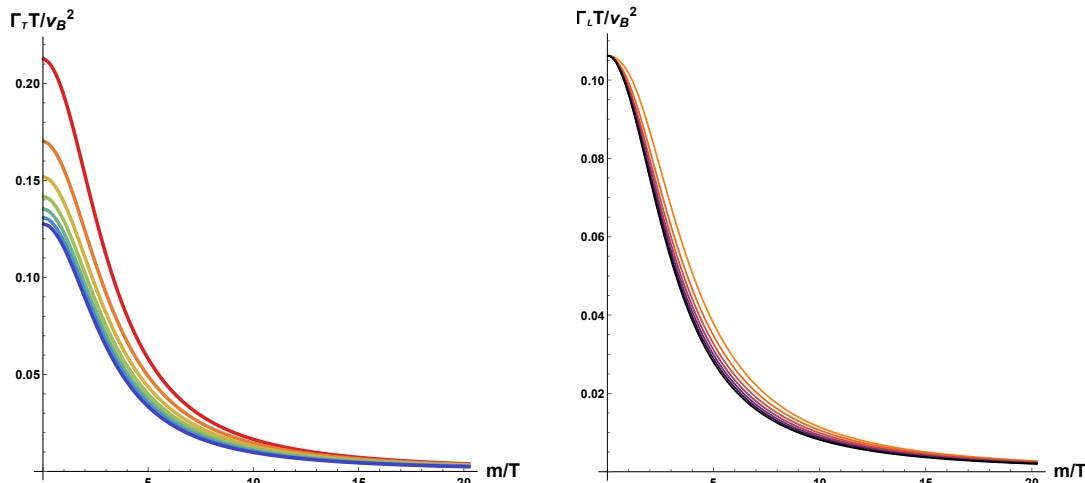

Figure 3: **Left:** The dimensionless ratio $\Gamma_T T/v_B^2$ in function of $m/T$ for various $N \in [3,9]$ from red to blue. **Right:** The dimensionless ratio $\Gamma_L T/v_B^2$ in function of $m/T$ for various $N \in [3,9]$ from orange to black.

conclusions can be achieved by considering anisotropic holographic models. In those models [58, 97, 98], the viscosity becomes a tensor and one component, the one parallel to the anisotropic direction, parametrically violates the KSS bound. Nevertheless, the diffusion constants keep obeying a lower bound, as shown in [28, 36]. Physically, this implies that the apparent violation of the KSS bound does not imply any dynamics "faster" than the Planckian scale. Morally, it confirms, once more, that $\eta/s$ is not the correct quantity to consider in a general context. To summarize, a universal lower bound on the momentum diffusion constant applies also to systems with broken translations and/or broken isotropy, in which the KSS bound is violated.

## 3.2 A bound on Goldstone diffusion

Let us move now to a different situation in which translations are broken only spontaneously. This is what happens in a solid with long-range order [68]. Within this scenario, the stress tensor is still a conserved operator and no momentum dissipation takes place. As a consequence, the hydrodynamic description does not break down at any value of the parameters and it continues to be valid as far as the dynamics of the Goldstone modes (the phonons in this case) is taken into account in the description. In this sense, a hydrodynamic description for ordered crystals was built long time ago in [99].

In the holographic models with SSB of translations (see for example [69]), the $\eta/s$ ratio is violated in the same way that happens in the previous cases [60, 100]. This raises a very interesting question. Why for the breaking of rotational invariance there is a neat difference between explicit [58] and spontaneous breaking [59] while in the case of momentum there is not? Can this be understood from a a scaling analysis like the one of [101]? Can we find a generalized bound taking into account the anisotropic scalings as discussed in [102, 103]?

Following the theory of elasticity as the effective field theory for spontaneously broken translations [68, 104–107], the hydrodynamic spectrum contains now transverse and longitu-

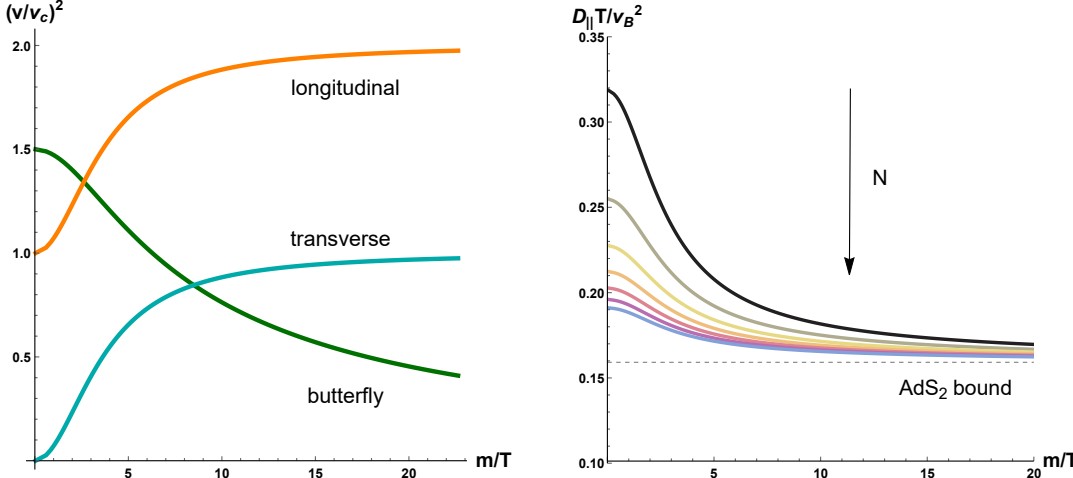

Figure 4: **Left:** The characteristic velocities in the holographic systems with SSB of translations. We fixed $N = 3$. In orange the speed of longitudinal sound; in cyan the speed of transverse sound and in green the butterfly velocity. All the speeds are normalized by the conformal value $v_c^2 = 1/2$. **Right:** The dimensionless ratio $D_\parallel T/v_B^2$ in function of $m/T$ for various $N \in [3, 9]$ (from black to blue). The dashed line is the AdS$_2$ universal value $1/2\pi$.

dinal propagating sound modes whose dispersion relation are respectively:

$$\omega_T = v_T\, k - \frac{i}{2}\, \Gamma_T\, k^2 + \dots, \tag{30}$$

$$\omega_L = v_L\, k - \frac{i}{2}\, \Gamma_L\, k^2 + \dots, \tag{31}$$

where this time the parameters $\Gamma_{T,L}$, controlled by the viscous coefficients (shear viscosity and eventually bulk viscosity), are the sound attenuation constants (and not the diffusion constants). Because of this simple reason, there is a priori no motivation to expect any universal bound applying to these quantities. Indeed, as shown in Fig.3, the dimensionless ratios:

$$\frac{\Gamma_{T,L}\, T}{v_B^2}, \tag{32}$$

where $v_B$ is the butterfly velocity [28, 29, 42, 51]:

$$v_B^2 = \frac{\pi\, T}{u_h}, \tag{33}$$

are not bounded by below and they go smoothly to zero value in the limit $m/T \to \infty$.

Nevertheless, in the holographic models under scrutiny, there is an additional diffusive mode in the longitudinal channel, which we will indicate as

$$\omega = -i\, D_\parallel\, k^2 + \mathcal{O}(k^4). \tag{34}$$

This diffusive mode comes from the spontaneous breaking of the global symmetry $\phi \to \phi + b$ [108] and its physical nature is still controversial. We refer the reader to [109] for a more in depth discussion on the subject and a possible physical interpretation in terms of quasicrystals physics. Despite the physics behind this mode remains obscure to date, its hydrodynamic

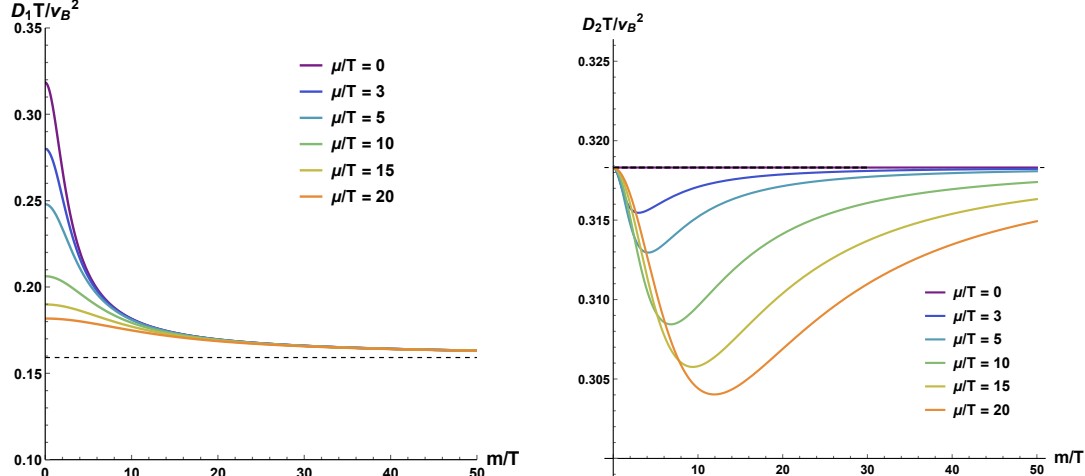

Figure 5: The longitudinal diffusion constants at finite charge. $D_1$ is the one which reduces to pure crystal diffusion at $\mu = 0$, while $D_2$ becomes the charge diffusion at $\mu = 0$. The dashed lines indicate the AdS$_2$ bounds: $1/2\pi$ (energy and crystal diffusion), $1/\pi$ (charge diffusion).

description [110] is well understood and was recently checked in [84]. In particular, at zero charge density, its diffusion constant is given by [110]:

$$D_\parallel = \xi \, \frac{(B + G - \mathcal{P}) \, \chi_{\pi\pi}}{s' \, T^2 \, v_L^2}, \tag{35}$$

where $\xi$ is the Goldstone dissipative parameter, $B$ the bulk modulus, $G$ the shear modulus, $\chi_{\pi\pi}$ the momentum susceptibility, $s'$ the temperature derivative of the entropy ($\sim$ the specific heat) and finally $\mathcal{P}$ the so called crystal pressure. The hydrodynamic formula above matches perfectly the numerical data obtained from the quasinormal modes [84].

We plot the dimensionless ratio $D_\parallel T / v_B^2$ in Fig.4. We find that:

$$\frac{D_\parallel \, T}{v_B^2} \geq \frac{1}{2\pi}, \tag{36}$$

where

$$\frac{1}{2\pi} \equiv \text{AdS}_2 \text{ bound for energy diffusion}, \tag{37}$$

as found in [35]. Importantly, we can verify numerically and analytically that the only diffusive mode present in our system with spontaneously broken translation respects the Hartnoll's bound [25], with the characteristic speed identified with the butterfly velocity, as proposed by Blake in [28, 29]. Given the presence of a several non-trivial speeds, one could wonder if this identification is the correct one and/or the only one working. It is easy to convince the reader that this is the case. In Fig.4, we plot the three different velocities appearing in our system. At large values of $m/T$, corresponding to very low temperatures or a very rigid/frozen system, the butterfly velocity behaves very differently from the phononic ones. More concretely, the speeds of sound reach a constant maximum at zero temperature, while the butterfly velocity decays to zero with a power law scaling:

$$v_B^2 \sim \left(\frac{m}{T}\right)^{-1}, \tag{38}$$

which is fundamental for the validity of the bound at low temperatures. As we will see later, the speed of sound, which grows towards small temperature, will act as an upper bound for the diffusion constants.

Using the results of [111], we can explore the dynamics of the diffusion mode even at finite charge density. In this case the problem is more complicated because the crystal diffusion mode couples directly to the charge diffusion mode, producing two longitudinal diffusive modes:

$$\omega_{1,2} = -i\,D_{1,2}\,k^2 + \mathcal{O}(k^4),\tag{39}$$

exactly as charge and thermal diffusions couple together in thermoelectric transport.
Despite knowing that in inhomogenous models or higher derivative models the universal bound on charge diffusion could be violated [50, 51], it is still interesting to see what happens to the two diffusive modes within this scenario. In order to do that, we have plotted the two dimensionless ratios $D_{1,2}\,T/v_B^2$ for several values of the chemical potential in Fig.5. Let us recall, that at zero chemical potential the two diffusion modes are decoupled and they correspond to:

$$D_1 = \text{crystal diffusion},\tag{40}$$
$$D_2 = \text{charge diffusion}.\tag{41}$$

At zero charge, the crystal diffusion mode is well above the AdS$_2$ bound for energy diffusion (37), which is saturated only at zero temperature, while the charge diffusion mode exactly saturates the AdS$_2$ bound for charge diffusion $\equiv 1/\pi$ [35]. When the coupling between the two diffusive modes is increased, the first diffusion constant slightly decreases without violating the universal bound. More interestingly, the second diffusive mode violates the bound for charge diffusion $\equiv 1/\pi$ [35]. Despite violations of this kind were already observed [51], here we do not need inhomogeneity or higher derivative couplings to achieve them. Interestingly enough, the violation is most severe at an intermediate value of $m/T$ and not in the limit $m/T \to \infty$.

Our analysis confirms that charge diffusion does not satisfy any universal bound in terms of the butterfly velocity. Nevertheless, the crystal diffusion mode does. Moreover, the zero temperature bound is exactly the same appearing for energy diffusion. It would be extremely interesting to understand (I) if the crystal diffusion bound is so universal as the energy diffusion one, (II) why, and (III) why energy diffusion and crystal diffusion display the same universal lower coefficient $\equiv 1/2\pi$.

### 3.3 An analytic check at zero charge density

In this section, we go beyond the numerical results and use some perturbative analytic methods to assess the validity of the bounds just discussed. The only transport coefficient for which we do not have an analytic formula at any value of $m/T$ is the shear modulus $G$, which comes from the decoupled equation for the $h_{xy}$ component of the metric. In particular, its definition in terms of Kubo formulas is given by:

$$G \equiv \text{Re}\,\mathcal{G}^{(R)}_{T_{xy}T_{xy}}(\omega = k = 0),\tag{42}$$

where $\mathcal{G}^{(R)}_{T_{xy}T_{xy}}$ is the retarded Green function for the $T_{xy}$ component of the stress tensor operator. It turns out that, at zero charge density and for a specific power of the potential $N = 3$, the equation becomes simple enough to be solved analytically.

More precisely, using the results of [80] expressed in Eddington-Filkenstein coordinates, the shear equation at zero frequency ($\omega = 0$) is given by:

$$\left(1-z^3\right)z\left((-\delta_m+3)z^3-3\right)h'' + 18(-\delta_m+3)z^5 h + \left(4(\delta_m-3)z^6 + (-\delta_m+6)z^3 + 6\right)h' = 0,\tag{43}$$

where $z \equiv u/u_h$, $h(z) \equiv \delta g_{xy}(z)$ and $\delta_m \equiv 3 - m^2$.

In these notations, the limit $m/T \to \infty$ corresponds to send $\delta_m \to 0$. It is simple to solve Eq.(43) at leading order in $\delta_m$, obtaining the general solution:

$$h(z) = \frac{c_1}{\left(z^3-1\right)^2} + \frac{c_2\left(z^6-3z^3+3\right)z^3}{9\left(z^3-1\right)^2}.\tag{44}$$

Imposing regularity at the horizon $z = 1$, we obtain the constraint $c_2 = -9\,c_1$. Finally, expanding the solution close to the boundary $z = 0$, we derive:

$$h(z) \sim 1 - u^3 + \ldots\tag{45}$$

This implies that the shear modulus in the limit of zero temperature is given by:

$$G^{(N=3)}_{m/T\to\infty} = \frac{3}{2}.\tag{46}$$

In Table 1, we summarize the values of the other relevant transport coefficients at zero charge and for $N = 3$ in the two limits $m/T \to 0$ and $m/T \to \infty$.

Using these values, we can derive analytically, at least for $N = 3$, that:

$$\lim_{m/T\to\infty} \frac{D_\parallel T}{v_B^2} = \frac{1}{2\pi}.\tag{47}$$

This is a nice confirmation that our numerics are trustful. Moreover, looking at the values listed in Table 1, it is clear that this bound arises from a very non-trivial cooperation of the various transport coefficients.

We do expect that using IR scaling arguments, one could actually generalize the bound for crystal diffusion to Hyperscaling and Lifshitz IR geometries, as done in the past for energy diffusion. Most likely, the only difference will appear in the numerical prefactor showing a non-trivial dependence on the Lifshitz-Hyperscaling parameters $z, \theta$.

## 4 An upper bound on diffusion

Let us now change completely point of view and ask ourselves how "fast" diffusion can go and what can limit it. This problem was analyzed in [15].

Consider a physical system where the existence of a lightcone defines a causal structure. Causal processes are those who occur inside of the lightcone and therefore they are not superluminal. In a relativistic theory, the lightcone speed is naturally given by the speed of light $c$. In non-relativistic systems that is not necessarily true. The lightcone speed has to be thought as a UV microscopic velocity; see more details about its definition in [15]. To proceed, let us assume the existence of a diffusive process obeying the Einstein-Stokes law:

$$\langle x^2 \rangle = D\,t.\tag{48}$$

Table 1: The values of the various thermodynamic and transport coefficients for $\mu = 0$ and $N = 3$. For simplicity we fixed $u_h = 1$ and we used the notation $\delta_m = 3 - m^2$.

| Coefficient | $m/T \to 0$ | $m/T \to \infty$ ($\delta_m \to 0$) |
|:---:|:---:|:---:|
| $T$ | $3/4\pi$ | $\delta_m/4\pi$ |
| $\chi$ | $3/2$ | $3$ |
| $G$ | $m^2$ | $3/2$ |
| $v_T^2$ | $0$ | $1/2$ |
| $v_L^2$ | $1/2$ | $1$ |
| $s'$ | $16\pi^2/3$ | $8\pi^2/9$ |
| $\mathcal{P}$ | $m^2$ | $3$ |
| $\mathcal{B}$ | $3 m^2$ | $9/2$ |
| $\xi$ | $1/(3 m^2)$ | $\delta_m^2/324$ |

In order for this diffusive dynamics to be causal, we need:

$$\sqrt{D\, t} \leq v_{\text{lightcone}}\, t\,, \tag{49}$$

which implies an upper bound $D \leq v_{\text{lightcone}}^2 t$. However, let us remind the reader that diffusion sets in only after the so-called equilibration or thermalization time $\tau_{eq}$, at which local equilibrium is reached . Therefore, assuming that diffusion begins at $t = \tau_{eq}$, we can derive a very general bound:

$$D \leq v_{\text{lightcone}}^2\, \tau_{eq}\,, \tag{50}$$

which is fundamentally implied by causality. A graphic representation of this constraint is provided in Fig.6.

At this point, the reader can notice that the inequality sign in (50) is reversed with respect to all our previous discussions. Interestingly, but somehow confusing, there appeared other instances where the bound on diffusion is an upper bound and not a lower bound. A detailed discussion can be found in [112] and a concrete example for a diffusion upper bound in [34]. Differently from the lower bound, this upper one can be easily understood in terms of fundamental constraints of the quantum theory. Can similar arguments apply to the lower bound? Is the diffusion constant, at the end of the day, constrained to live in an intermediate range:

$$v_B^2 \tau_{pl} \lesssim D \lesssim v_{ligthcone}^2\, \tau_{eq}\,, \tag{51}$$

where $\tau_{eq}$ is the thermalization (or equilibration) time at which diffusion sets in?

The equilibration time is generally longer than the Planckian one, which is believed to be the minimum value available in nature. This argument relates directly to a universal bound for relaxation, discussed in black hole physics [113] and also in the context of the weak gravity conjecture [114]. Nevertheless, when the two timescales coincide, and assuming the butterfly velocity determines the lightcone speed, the window for diffusion shrinks to a point and set the diffusion constant to be $D \simeq v_B^2 \tau_{pl}$. Can we understand the saturation of the bound as the limit in which the equilibration time reaches its Planckian minimum? Can we rationalize the cases, such as [34], where it was found $D \leq v_B^2 \tau_{pl}$?

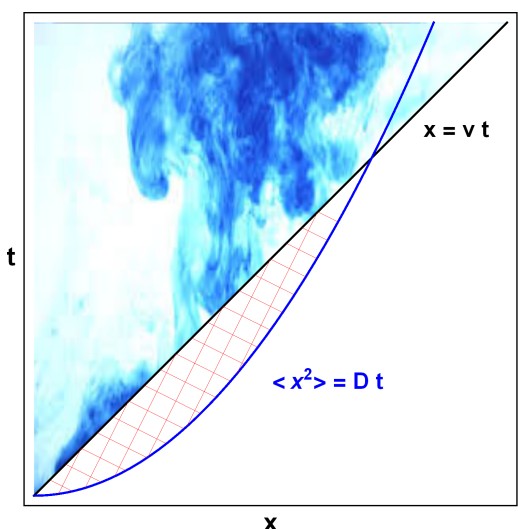

Figure 6: A simple visual derivation of the upper bound on diffusion imposed by causality. $v$ here is the lightcone speed. Figure adapted from [15].

## 4.1 Causality of Relativistic Hydrodynamics as a diffusivity bound

Let us start by discussing a simple situation where this bound is manifested: relativistic hydrodynamics. Relativistic hydrodynamics [64], in absence of additional U(1) charges, is entirely governed by the conservation of the stress tensor:

$$\nabla_\mu T^{\mu\nu} = 0\,, \tag{52}$$

and by the additional constitutive relation which at leading order takes the form

$$T^{\mu\nu} = \epsilon\, v^\mu v^\nu + p\,\Delta^{\mu\nu} - \eta\,\sigma^{\mu\nu}\,, \tag{53}$$

where $\eta, \epsilon, p$ are respectively the shear viscosity, the energy density and the pressure. $v^\mu$ is the four velocity, $\Delta^{\mu\nu}$ the standard projector and $\sigma^{\mu\nu}$ the strain rate – the time derivative of the stain tensor. Using standard hydrodynamics techniques, it is simple to show that the dynamics in the transverse sector is governed by a shear diffusive mode

$$\omega = -i\,\frac{\eta}{\epsilon + p}\,k^2\,, \tag{54}$$

which originates from the conservation of momentum.

It is a known problem, that linearized relativistic hydrodynamics, as just described, is acausal because the group velocity of the shear mode can become superluminal

$$|v| = \left|\frac{\partial \omega}{\partial k}\right| \sim k > c\,, \tag{55}$$

where large momenta are considered. Despite, this is certainly not a fundamental problem of the theoretical description, it is an issue for the hydrodynamic numerical simulations. One famous resolution, known as Israel-Stewart formalism [115], consists in expanding the stress tensor constitutive relation (53) to higher orders, introducing the fictitious relaxation time $\tau_\pi$. Doing that, the dispersion relation of the diffusive shear mode is now modified. More precisely, it is given as a solution of

$$\omega^2 + i\,\omega\,\tau_\pi^{-1} = v^2 k^2\,, \qquad v^2 = \frac{\eta}{(\epsilon + p)\,\tau_\pi}\,, \tag{56}$$

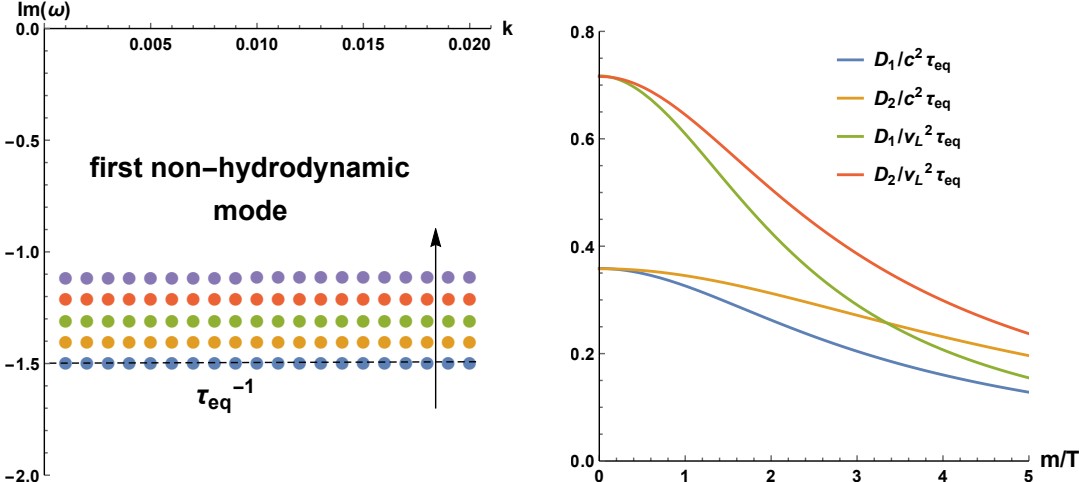

Figure 7: The upper bond on diffusion at zero charge. In this case $D_1$ is the crystal diffusion and $D_2$ the charge diffusion. **Left:** We extracted the thermalization time from the imaginary part of the first non-hydrodynamic mode as in Eq.(60). $c$ is the speed of light and $v_L$ is the speed of longitudinal sound. The arrow in the upper panel indicates the motion increasing $m/T$. **Right:** The dimensionless ratios. All of them get values in the range $[0, 1]$, implying that the upper bound is respected.

which is usually referred to as the *telegraph equation*. This equation has been recently extensively discussed in the context of liquids dynamics and it gives rise to the so-called gapped momentum states [116], which have been observed in holographic models [79, 117]. This relation, which can be obtained using quasi-hydrodynamics [118] and field theory for dissipative systems [119], implies that the shear diffusive mode (54) survives only up to a certain cutoff momentum:

$$k_g \equiv \frac{1}{v\,\tau_\pi}. \tag{57}$$

Above this momentum cutoff, a propagating mode with asymptotic speed $v$ appears. In order to ensure that the speed of such mode is subluminal, we impose:

$$v < c \quad \rightarrow \quad \sqrt{\frac{D}{\tau_\pi}} < c\,. \tag{58}$$

Interestingly enough, the previous expression can be re-written in the form of a diffusive bound (50) as

$$D \leq c^2\,\tau_\pi, \tag{59}$$

where the lightcone speed is given by the speed of light $c$ and the relaxation time by the Israel-Stewart timescale. In other words, the causality of the Israel-Stewart theory can be understood as an upper bound on the shear diffusion constant. This is appealing since it connects a transport bound to a fundamental requirement of the underlying theory such as causality.

## 4.2 Results from holography

A relevant question is how to define the equilibration time in general setups, and in particular within holography. A possibility is to use the imaginary part of the first non-hydrodynamic mode:

$$\tau_{eq}^{-1} \equiv \operatorname{Im}\omega_{\text{QNM}}, \tag{60}$$

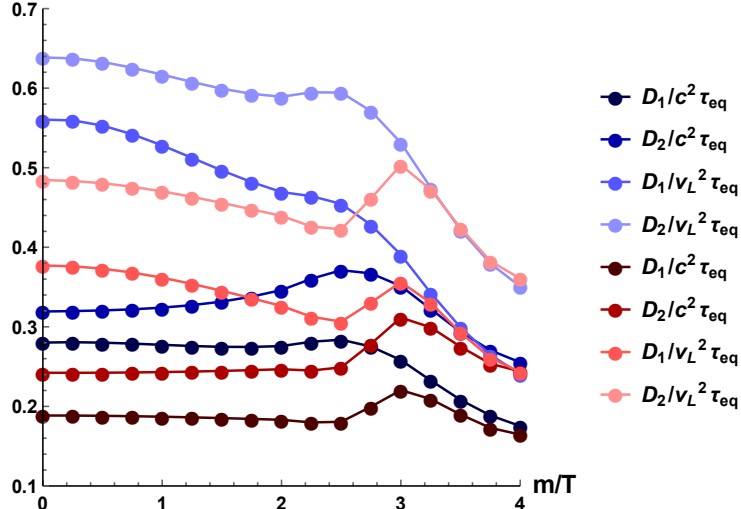

Figure 8: A verification of the universal upper bound (50) imposed by causality. The blue points are for $\mu/T = 3$ while the red for $\mu/T = 5$.

which sets the smallest dissipative scale. At first sight, it looks odd to expect any universal physics coming from non-hydrodynamic gapped modes. A more modern and fundamental approach suggests that the local equilibration time $\tau_{eq}$ has to be identified with the position of the so-called *critical points* [120]. The latter are related to the regime of applicability of hydrodynamics and can be found by looking at the positions at which the polynomial $P(k, \omega)$ (whose zeros give the QNMs of the system) and its derivatives vanish. In general, this position can be different from that of the first gapped mode. Nevertheless, in the cases which were explicitly checked [79], we did not find more than an $\mathcal{O}(1)$ variation. It would be interesting to study in more detail these points in our class of models.

We investigate the validity of the upper bound Eq.(50) in a large class of holographic models with spontaneously broken translations. We start with zero charge density, $\rho = 0$. It is simple to identify numerically the first non-hydrodynamic mode in the spectrum, which is shown in the left panel of Figure 7 for different values of $m/T$ (which in this case sets the rigidity of the system). By extracting the relaxation time $\tau_{eq}$ numerically, we can test the upper bound on diffusion (50). In the right panel of Fig.7, we observe that the bound is universally obeyed using both the speed of light, but even the speed of longitudinal sound. We generically find that:

$$D_{1,2} \leq \mathfrak{N} v_L^2 \tau_{eq}, \tag{61}$$

where $D_1$ is the crystal diffusion and $D_2$ charge diffusion and $\mathfrak{N}$ a pure number of order $\mathcal{O}(1)$. Finally, we check the universal upper bound (50) at finite charge density, at which the two longitudinal diffusive modes are mixing. In Fig.8 we show some benchmark results at $\mu/T = 3, 5$. We conclude by confirming that any diffusive process in our systems with broken translations obeys the upper bound set by causality.

## 5 Bounds on the speed of sound and the stiffness

In this section, we slightly change our gear and move to the possibility of bounding propagating modes and in particular their speeds of propagation. In [65, 66] it was conjectured that any field theory with a gauge/gravity dual can have a speed of sound at most as large as that of a

conformal theory:

$$v_c^2 \equiv \frac{1}{d-1} c^2, \tag{62}$$

with $d$ the number of spacetime dimensions.

This proposal was later related to the physics of neutron stars, very compact objects which display an extremely stiff equation of state [121]. The authors of [122] found that this bound could be violated at finite charge and later Ref. [123] showed the violation even at zero charge by adding multitrace deformations.

Importantly, all the previous discussions focused on systems with no long-range order – fluids – for which:

$$v_s^2 = \frac{\partial p}{\partial \epsilon}. \tag{63}$$

This implies that in that scenario the upper bound on the speed of sound was simultaneously an upper bound on the stiffness of the equation of states $\kappa \equiv \frac{\partial p}{\partial \epsilon}$.

Nevertheless, the sound that we hear when we knock the table does not come from such dynamics. On the contrary, it is well known that, in most of the systems in nature, sound relates to rigidity and the mechanical vibrations of the material – phonons. It is therefore interesting to understand if the speed of sound, intended as the speed of propagation of longitudinal phonons, is universally bounded from above and by what. Recently, Ref. [10] proposed a bound coming from the so-called melting temperature:

$$v_s \le v_m \equiv \frac{k_B T_m a}{\hbar}, \tag{64}$$

which plays a fundamental role on the validity of the Planckian bound for high temperature thermal transport in insulators. There, $T_m$ is the melting temperature of the material and $a$ the lattice spacing. Notice, that the bound in (64) involves microscopic quantities, which are in general extremely dependent on the structural details of the solid. It is therefore not so clear how to make such bound more universal, in the sense of agnostic to any microscopic features. Additionally, the authors of [67] suggested a different, but closely related, bound on the speed of sound written in terms of fundamental physical constants:

$$v_s \le \alpha \left( \frac{m_e}{2 m_p} \right)^{1/2} c, \tag{65}$$

and obeyed by a large set of experimental data and *ab-initio* computations for atomic hydrogen. Here, $\alpha$ is the fine structure constant and $m_e, m_p$ respectively the electron mass and the proton mass.

Given the large amount of work related to holographic solids and phonons, it is an extremely interesting question to see if the sound speed therein obeys any sort of upper bound. As a starting point, we will investigate the sound speed in a large class of holographic conformal models for which:

$$v_L^2 = \frac{1}{2} c^2 + v_T^2. \tag{66}$$

This very interesting relation, found in [124], and discussed later in [106], just follows from the vanishing of the stress tensor trace:

$$T^\mu{}_\mu = 0. \tag{67}$$

One could obviously break this assumption by introducing for example a non-trivial dilaton field in the bulk and relaxing conformal invariance.

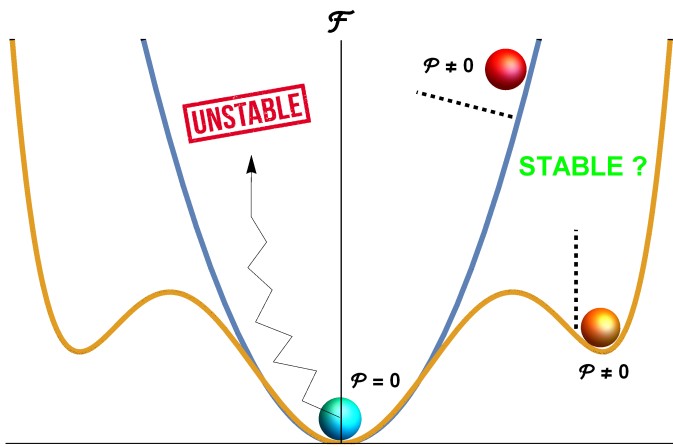

Figure 9: The puzzle of stability for the homogeneous holographic models (axions-like and Q-lattices like) with SSB of translations. The thermodynamically favoured phases with $\mathcal{P} = 0$, which minimize the free energy $\mathcal{F}$, are all unstable! On the contrary, all the phases which do not minimize the free energy do not present any dynamical instability (at least at linear level). Are those phases metastable? Where does the instability of the favoured phases come from?

Notice that, taking $N < 3/2$, the shear modulus $G$ would clearly be negative, leading to a dynamically unstable model. This can be immediately inferred by looking at the analytic formula for the shear modulus $G$:

$$G = \frac{N}{2N-3} m^2 u_h^{2N-3} + \mathcal{O}(m^4),\tag{68}$$

which was presented in [69] and which is valid at leading order in $m/T$. This dynamical gradient instability is indeed present in the holographic model of [110,125] and in the thermodynamically stable (but dynamically unstable!) models of [84,126,127]. Importantly, these classes of homogeneous models (axions-like and Q-lattices) present us an interesting puzzle (see Fig.9). All the models with finite lattice pressure $\mathcal{P}$ do not minimize the free energy but they do not present any dynamical instability. On the contrary, the thermodynamic favoured models, with $\mathcal{P} = 0$, are all unstable and they should be discarded! This observation opens an interesting set of questions: (a) are the homogeneous models with finite crystal pressure metastable? Can any instability appear in those models with $\mathcal{P} \neq 0$ if one goes beyond linear response? (b) Why are all the thermodynamically stable models dynamically unstable? Is there any fundamental physics behind? Preliminary observations about metastability and these global × spacetime symmetry breaking patterns have already appeared in [128,129]. It would be interesting to extend them at finite temperature.

To ensure dynamical stability, we consider only $N > 3/2$. In Fig.10, we plot the longitudinal speed of sound at various temperature by moving the power $N$ in the potential. As already noticed and discussed in depth in [106], we observe that for $N < 3$ the speed of sound is superluminal. It is not clear how serious this superluminal violation should be considered, since the (emergent) lightcone speed can in principle be different from the speed of light $c$. This is for example the case in graphene [130], where the lightcone speed is $\approx c/300$ but the 3-dimensional photons move with a speed larger than the lightcone one. Interestingly, if we ignore the constraint of superluminality, we observe a maximum in the speed of sound which is given by

$$v_L^2 \leq \frac{\pi^2}{8} v_{lightcone}^2.\tag{69}$$

At this stage, do not have a clear understanding of where this maximum value comes from. Methods similar to those of [131] could shed light on this maximum value emerging in our class of models. Moreover, we notice that the speed of sound always exceeds the conformal value:

$$v_L^2 \geq \frac{1}{2}. \tag{70}$$

This conclusion can be immediately derived from the fact that:

$$v_L^2 = \frac{1}{2} + \frac{G}{\chi_{\pi\pi}}, \tag{71}$$

with the shear modulus $G > 0$

We conclude, that in (conformal) solids there is no reason to expect the speed of sound being limited by the conformal value. Additionally, it can be proved explicitly that the conformal value in this case acts as a lower bound:

$$v_L^2 \geq v_c^2. \tag{72}$$

For completeness, let us investigate what happens to the stiffness $\kappa \equiv \partial p / \partial \epsilon$ in these holographic conformal systems. First, it is important to notice that because of the presence of a finite crystal diffusion $\mathcal{P}$, the stiffness defined via the thermodynamic pressure does not show the usual conformal value, even if conformal symmetry is preserved. On the contrary:

$$\langle T^{xx} \rangle = \frac{1}{2}\epsilon, \tag{73}$$

together with:

$$\langle T^{xx} \rangle = p + \mathcal{P}, \qquad \mathcal{P} > 0. \tag{74}$$

From these simple arguments, we can immediately obtain that:

$$\kappa \equiv \frac{\partial p}{\partial \epsilon} = \frac{1}{2} - \frac{\partial \mathcal{P}}{\partial \epsilon}. \tag{75}$$

Assuming that $\frac{\partial \mathcal{P}}{\partial \epsilon} > 0$, we can confirm that there is indeed an upper bound on the stiffness of the system:

$$\kappa \leq \kappa_c, \qquad \kappa_c \equiv \frac{1}{d-1}, \tag{76}$$

but not on the speed of sound!

To confirm this result, we compute analytically the stiffness of our system using:

$$\mathcal{P} = \frac{m^2 N u_h^{2N}}{(2N-3)u_h^3}, \qquad \epsilon = \frac{1}{u_h^3} - \frac{m^2 u_h^{2N}}{(3-2N)u_h^3}. \tag{77}$$

From these expressions, it is simple to check that $\partial \mathcal{P} / \partial \epsilon$ is positive for all $N > 3/2$, namely for all those models where translations are broken spontaneously and there is a well-defined propagating sound mode with a positive shear modulus $G > 0$. The situation is different in the models with smaller $N < 3/2$, where the shear modulus becomes negative. In those cases, it seems that the stiffness can violate the conformal bound and even become negative! This feature appears very odd and analogous to the fact that for $N < 3/2$ the energy density (and also the crystal pressure) can become negative. Notice however that the stiffness being negative does not a priori implies any dynamical instability. The reason is simply that for $N < 3/2$ momentum dissipation destroys the propagating sound mode and there is no mode in the spectrum with speed given by the stiffness. This has been verified explicitly for $N = 1$ in [132].

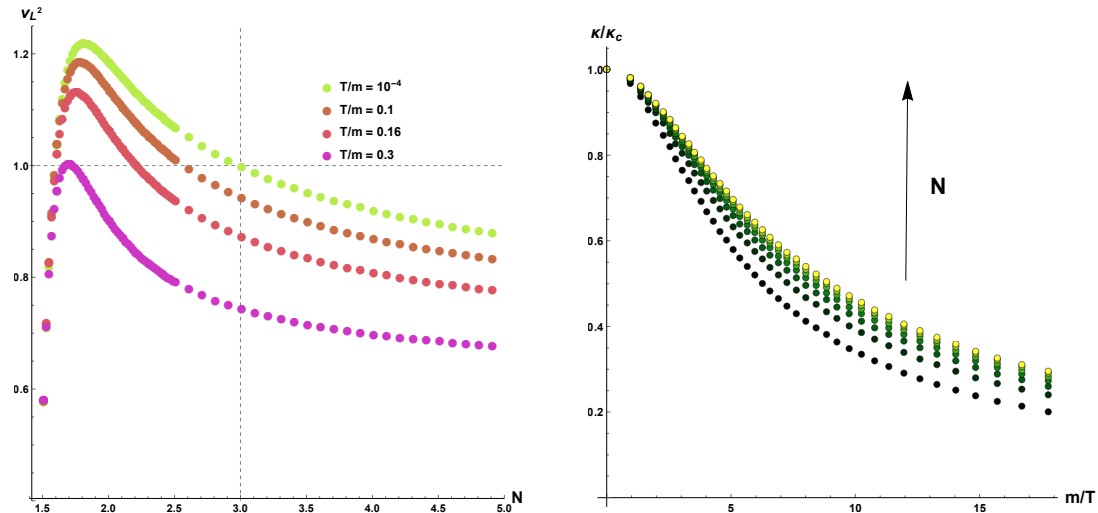

Figure 10: **Left:** The speed of longitudinal sound $v_L^2$ at different temperatures in function of $N$. We constrained $N > 3/2$ because below that value $G < 0$ and the models are dynamically unstable. **Right:** The stiffness $\kappa \equiv \partial p / \partial \epsilon$ for various $N \in [3, 9]$ (from black to yellow) in function of $m/T$.

Finally, we plot the stiffness $\kappa$ in Fig.10 for the models with spontaneously broken translations. As expected, the conformal value acts as a universal upper bound, confirming our expectations. In conclusion, in these systems, the conformal value for the speed of sound acts as a lower bound but the stiffness is still bounded by above by the conformal value. This implies tha the bounds conjectured in [65,66] has to be thought as limits to the stiffness and not as bounds on the speed of sound[2], which can easily be larger than the conformal value.

To conclude, let us remark that the holographic models considered in this work presents several important drawbacks (compared to "real solids") : (I) they are conformal; (II) they lack a UV cutoff playing the role of lattice spacing in real crystals and (III) they don't present any first order melting transition. It is definitely worth to generalize them and see how many of our results depend on those assumptions. Relaxing conformal symmetry, it appears difficult to find a universal (lower or upper) bound on the speed of sound without resorting to any UV physics controlled by the microscopics of the systems such as the lattice spacing.

## 6   Conclusions

We conclude with a brief summary of our results and a few comments and open questions. First, we have shown numerically that the violation of the KSS bound in holographic systems with momentum dissipation can be saved by looking at the momentum diffusivity, as a more natural quantity to bound. This is totally consistent with the lesson of [25] and with the two observations that the $\eta/s$ ratio is not well-defined in non-relativistic systems and furthermore it does not control any transport mechanism in more general setups.

Second, we have shown that in holographic models with spontaneously broken translations, in which the shear mode is propagating and not diffusive, the diffusion constant of the

---

[2]We thank Carlos Hoyos for this suggestion.

longitudinal Goldstone diffusion mode always obeys a universal bound of the type:

$$D_\parallel \geq \frac{1}{2\pi} v_B^2 \tau_{pl}. \tag{78}$$

Third, we have checked numerically that all the diffusive processes in the holographic models with broken translations considered obey the upper bound proposed in [15]. The lightcone speed appearing in such bound can be taken to be the speed of longitudinal phonons, while the thermalization time can be identified with the imaginary part of the first non-hydrodynamic mode present in the spectrum.

Finally, we have shown that, differently from what conjectured in the past [65, 66], in holographic (rigid) systems with finite elastic moduli, the conformal value acts as a lower bound for the longitudinal speed of sound and not as an upper one. Nevertheless, the stiffness $\partial p/\partial \epsilon$, is indeed bounded by above by its conformal value.

In conclusions, we summarize all the results presented and discussed in the present work in Table 2.

Several are the questions still open. We have already presented a lot of them through the main text. Here, we conclude by adding a few more.

The first direct question relates to the universality of the bound for the crystal diffusion mode. In particular it would be interesting to understand its physical implications and its relation (if any) with energy diffusion.

In general, it would be very profitable to identify a clear way to introduce a UV cutoff in the holographic models for solids or a scale related to the lattice spacing. Several are the properties of solids connected to the lattice spacing such as melting, the saturation of the specific heat at large temperature (Dulong-Petit law) and the presence of Van Hove singularities. Moreover, both the bounds on the speed of sound proposed in [10, 67] rely on the existence of such UV scale. An interesting model in this direction is that of [133], which presents a UV regulator appearing in the renormalization procedure and affecting the dynamics of the phonons.

Another fundamental aspect is the possibility of defining a melting temperature and test holographically the bound of [10]. A scenario in which the melting mechanism is discussed is that of [124], where the melting temperature has been associated to the Hawking-Page transition in the bulk. The Hawking-Page transition in holographic models with massive gravitons has been already partially analyzed in [134]; it would be interesting to extend their computations to the holographic models for solids considered in this work.

Moreover, one could ask if hydrodynamic fluctuations [135] lead to the violation of the diffusivity bounds and more generically how do they affect them.

Finally, it would be worth considering if universal bounds on transport (e.g. scalings, exponents, etc ...) can exist beyond linear response. One direct option would be to look at the non-linear stress-strain curves as discussed in [136, 137].

More broadly, we end it here by reminding the reader that holography and hydrodynamics are perfect platforms to study the existence of universal bounds in collective transports. In the near future, we hope to push both frameworks further towards their bounds.

Table 2: A summary of our results. EXB stands for explicit breaking of translations and SSB for spontaneous breaking of translations. The green, red and black symbols indicate respectively the validity, the violation and the inapplicability of the corresponding bound.

| Bound | References | Formula | EXB | SSB |
|---|---|---|---|---|
| KSS bound | [8] | $\frac{\eta}{s} \geq \frac{1}{4\pi}\frac{\hbar}{k_B}$ | ✗ [60, 61] | ✗ [69] |
| momentum diffusion | [96] | $D_T\,T \geq \frac{1}{4\pi}$ | ✓ [this paper] | ⊘ |
| crystal diffusion | [25, 28, 29] | $D_\phi \geq \# v_B^2\,\tau_{pl}$ | ⊘ | ✓ [this paper] |
| charge diffusion | [25, 28, 29] | $D_q \geq \# v_B^2\,\tau_{pl}$ | ✓ | ✗ [this paper] |
| causality bound | [15] | $D \leq v_{lightcone}^2\,\tau_{eq}$ | ✓ [this paper] | ✓ [this paper] |
| sound speed | [65, 66] | $v_L^2 \geq v_c^2 \equiv \frac{1}{d-1}$ | ⊘ | ✗ [this paper] |
| stiffness | [65, 66] | $\frac{\partial p}{\partial \epsilon} \geq \kappa_c \equiv \frac{1}{d-1}$ | ✗ [this paper] | ✓ [this paper] |

# Acknowledgements

We thank Y.Ahn, K.Benhia, L.Delacretaz, S.Grozdanov, S.Hartnoll, C.Hoyos, A.Lucas, H.-S.Jeong, K.-Y.Kim, L.Li, N.Poovuttikul, K.Trachenko and J.Zaanen for useful discussions and comments. We thank A.Jimenez and S.Grieninger for kindly providing the numerical codes used to produce the data in this paper. M.B. acknowledges the support of the Spanish MINECO "Centro de Excelencia Severo Ochoa" Programme under grant SEV-2012-0249. W.J.L. is supported by NFSC Grant No. 11905024 and DUT19LK20.

# A    Technical details

In this appendix we provide more details about the computations performed in this work and the methods used.

## A.1    Transverse sector

We assume the momentum $\vec{k}$ to be aligned along the $y$ direction. Using these notations, the set of perturbations in the transverse spectrum is given by:

$$a_x,\ h_{tx} \equiv u^2 \delta g_{tx},\ h_{xy} \equiv u^2 \delta g_{xy},\ \delta\phi_x,\ \delta g_{xu}. \tag{79}$$

For simplicity, we assume the radial gauge, *i.e.* $\delta g_{xu} = 0$.
The corresponding equations of motion in terms of the background line-element (14) are:

$$
\begin{aligned}
0 = {}& -2(1 - u^2\ddot{V}/\dot{V})h_{tx} + u\,h'_{tx} - i\,k\,u\,h_{xy} - \left(k^2 u + 2\,i\,\omega(1 - u^2\ddot{V}/\dot{V})\right)\delta\phi_x + u\,f\,\delta\phi''_x \\
& + \left(-2(1 - u^2\ddot{V}/\dot{V})f + u(2i\omega + f')\right)\delta\phi'_x \; ; \\[4pt]
0 = {}& 2\,i\,m^2 u^2\omega\dot{V}\,\delta\phi_x + u^2 k\,\omega\,h_{xy} + 2u^4\mu(i\omega a_x + f\,a'_x) \\
& + \left(6 + k^2 u^2 - 4u^2\mu^2 - 2m^2(V - u^2\dot{V}) - 6f + 2uf'\right)h_{tx} + \left(2uf - iu^2\omega\right)h'_{tx} - u^2 f\,h''_{tx} \; ; \\[4pt]
0 = {}& 2i\,k\,u\,h_{tx} - iku^2 h'_{tx} - 2i\,k\,m^2 u^2\dot{V}\,\delta\phi_x + 2h_{xy}\left(3 + i\,u\,\omega - 3f + uf' - m^2(V - u^2\dot{V})\right) \\
& - \left(2i\,u^2\,\omega - 2uf + u^2 f'\right)h'_{xy} - u^2 f\,h''_{xy} \; ; \\[4pt]
0 = {}& 2h'_{tx} - u\,h''_{tx} - 2m^2 u\,\dot{V}\,\delta\phi'_x + ik\,u\,h'_{xy} + 2u^4\mu\,a'_x \; ; \\[4pt]
0 = {}& -\mu\,h'_{tx} - k^2 a_x + (2i\,\omega + f')a'_x + f\,a''_x \,,
\end{aligned}
\tag{80}
$$

where $' \equiv \partial_u$ and

$$
\dot{V} \equiv \frac{dV(X)}{dX}, \qquad \ddot{V} \equiv \frac{d^2 V(X)}{dX^2} \,.
\tag{81}
$$

The UV asymptotics for the perturbations considered are given by:

$$
\begin{aligned}
\delta\phi_x &= \phi_{x(l)}(1 + \ldots) + \phi_{x(s)}u^{5-2n}(1 + \ldots), \\
h_{tx} &= h_{tx(l)}(1 + \ldots) + h_{tx(s)}u^3(1 + \ldots), \\
h_{xy} &= h_{xy(l)}(1 + \ldots) + h_{xy(s)}u^3(1 + \ldots), \\
a_x &= a_{x(l)}(1 + \ldots) + a_{x(s)}u(1 + \ldots) \,,
\end{aligned}
\tag{82}
$$

where the boundary is located at $u = 0$.
We solve numerically the previous set of equations and we extract the QNMs and the Green function for the stress operator $T_{xy}$ defined as

$$
\mathcal{G}^{(R)}_{T_{xy}T_{xy}} = \frac{2\Delta - d}{2}\frac{h_{xy(s)}}{h_{xy(l)}} = \frac{3}{2}\frac{h_{xy(s)}}{h_{xy(l)}} \,,
\tag{83}
$$

where the time and space dependences are omitted for simplicity.

## A.2 Longitudinal sector

The analysis of the longitudinal sector is totally analogous to that described for the transverse sector in the previous section. In this case, the perturbations are:

$$
\{h_{x,s} = 1/2\,(h_{xx} + h_{yy}),\; h_{x,a} = 1/2\,(h_{xx} - h_{yy}),\; \delta\phi_y,\; h_{tt},\; h_{ty}\} \,,
\tag{84}
$$

where we again assumed the radial gauge.

The final set of equations reads

$$
u f' \delta\phi'_y \dot{V} + 2 u^2 f \delta\phi'_y \ddot{V} + u f \delta\phi''_y \dot{V} - 2 f \delta\phi'_y \dot{V} - k^2 u \delta\phi_y \dot{V} - k^2 u^3 \delta\phi_y \ddot{V}
$$
$$
+ i k u h_{x,a} \dot{V} - i k u^3 h_{x,s} \ddot{V} + 2 i u^2 \omega \delta\phi_y \ddot{V} + u h'_{ty} \dot{V} + 2 i u \omega \delta\phi'_y \dot{V}
$$
$$
- 2 i \omega \delta\phi_y \dot{V} - 2 h_{ty} \left( \dot{V} - u^2 \ddot{V} \right) = 0 \tag{85}
$$

$$
u(f \left( u f' h'_{x,s} - 2 f h''_{x,s} - 2 i k m^2 u^3 \delta\phi_y \ddot{V} - u h''_{tt} + 4 h'_{tt} \right)
$$
$$
+ k h_{ty} \left( i u f' - 2 i f + 2 u \omega \right) + h_{x,s} \left( 2 m^2 u^3 f \ddot{V} + \omega \left( i u f' - 2 i f + 2 u \omega \right) \right))
$$
$$
+ h_{tt} \left( u \left( -u f'' + 4 f' + 2 m^2 u \dot{V} \right) - 12 f + k^2 u^2 - 2 m^2 V - 2 i u \omega + 6 \right) = 0 \tag{86}
$$

$$
2 h_{ty} \left( u \left( f' + m^2 u \dot{V} \right) - 3 f - m^2 V + 3 \right)
$$
$$
- u \left( u f h''_{ty} - 2 f h'_{ty} + i k u h'_{tt} + k u \omega h_{x,s} + k u \omega h_{x,a} - 2 i m^2 u \omega \delta\phi_y \dot{V} + i u \omega h'_{ty} \right)
$$
$$
+ 2 i k u h_{tt} = 0 \tag{87}
$$

$$
h_{x,s} \left( 2 u \left( f' + m^2 u \dot{V} \right) - 6 f + k^2 u^2 - 2 m^2 V + 4 i u \omega + 6 \right) - u^2 f' h'_{x,s}
$$
$$
- u^2 f' h'_{x,a} + 2 u h_{x,a} f' - u^2 f h''_{x,s} - u^2 f h''_{x,a} + 4 u f h'_{x,s} + 2 u f h'_{x,a} - 6 f h_{x,a}
$$
$$
+ k^2 u^2 h_{x,a} + 2 i k u h_{ty} + 2 m^2 u^2 h_{x,a} \dot{V} - 2 m^2 h_{x,a} V - 2 i u^2 \omega h'_{x,s}
$$
$$
- 2 i u^2 \omega h'_{x,a} - 2 u h'_{tt} + 6 h_{tt}(u) + 2 i u \omega h_{x,a} + 6 h_{x,a} = 0 \tag{88}
$$

$$
h_{x,s} \left( 2 u \left( f' + m^2 u \dot{V} \right) - 6 f + k^2 u^2 - 2 m^2 V + 4 i u \omega + 6 \right) - u^2 f' h'_{x,s} + u^2 f' h'_{x,a}
$$
$$
- 2 u h_{x,a} f' - u^2 f h''_{x,s} + u^2 f h''_{x,a} + 4 u f h'_{x,s} - 2 u f h'_{x,a} + 6 f h_{x,a} + k^2 u^2 h_{x,a}
$$
$$
- 4 i k m^2 u^2 \delta\phi_y \dot{V} - 2 i k u^2 h'_{ty} + 6 i k u h_{ty} - 2 m^2 u^2 h_{x,a} \dot{V} + 2 m^2 h_{x,a} V
$$
$$
- 2 i u^2 \omega h'_{x,s} + 2 i u^2 \omega h'_{x,a} - 2 u h'_{tt} + 6 h_{tt} - 2 i u \omega h_{x,a} - 6 h_{x,a} = 0 \tag{89}
$$

$$
- 6 h_{tt} + u (u f' h'_{x,s} - 2 f h'_{x,s} - 2 i k m^2 u^3 \delta\phi_y \ddot{V} + i k u h'_{ty} - 2 i k h_{ty}
$$
$$
+ 2 m^2 u^3 h_{x,s} \ddot{V} - u h''_{tt} + 4 h'_{tt} + 2 i u \omega h'_{x,s} - 2 i \omega h_{x,s}) = 0 \tag{90}
$$

$$
k u \left( h'_{x,s} + h'_{x,a} \right) - i u \left( 2 m^2 \delta\phi'_y \dot{V} + h''_{ty} \right) + 2 i h'_{ty} = 0 \tag{91}
$$

$$
h''_{x,s} = 0, \tag{92}
$$

where we have used the same notations $\dot{V} \equiv dV(X)/dX$, $\ddot{V} \equiv d^2 V(X)/dX^2$.

## A.3 Numerical method

In both sectors we have used pseudospectal techniques to determine the quasi-normal modes of the system. The fields are decomposed into Chebyshev polynomials, which ensure regularity at the horizon and automatically force the leading, divergent, terms of the asymptotic expansion to vanish. In this way, the problem of finding the QNMs is reduced to a generalized eigenvalue problem which displays several advantages with respect to the standard shooting methods. All our data have been obtained using 50 gridpoints and 60 digits precision. The convergence of the method has been verified numerically.

## A.4 Diffusion constants

In this last appendix, we discuss very briefly how the diffusion constants shown in the main text are derived.

We compute numerically the spectrum of the QNMs and we find numerically the dynamics of the eigenfrequencies in function of the momentum $k$. That allows us to define numerically a dispersion relation $\omega(k)$. Then, given a diffusive dispersion relation $\omega = -iDk^2$, valid at low

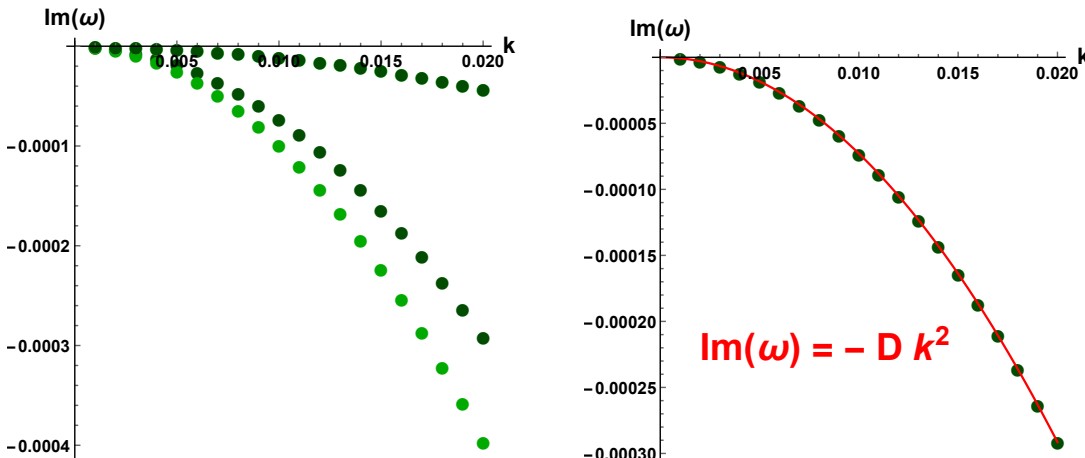

Figure 11: One example of the imaginary part of the QNMs spectrum (**left**) and the fitting done to extract the diffusion constant $D$ from a diffusive mode (**right**).

momentum, we numerically fit the data and we extract the value of the diffusion constant $D$ from the imaginary part of it. An example of this procedure is shown in Fig.11. Fortunately, given the results of [84, 111], we dispose of analytic hydrodynamic formulas for the various diffusion constants given in terms of the various transport coefficients. Therefore, we do not need anymore to compute the finite momentum QNMs but we can just use them. Those appear to be much easier given that most of the transport coefficients have analytic formulas and the rest of them can be derived with simple Kubo formulas at zero momentum. We refer to [84, 111] for more details.

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
