# Peer review of "Universal Bounds on Transport in Holographic Systems with Broken Translations"

_SciPost Physics, doi:SciPost Phys. 9, 007 (2020)_

## Round 1 · Referee Report · Anonymous (Referee 1) · 2020-6-3

Report

The authors study the behaviour of black hole solutions in a specific class of models corresponding to massive gravity theories in order to test the validity of several conjectured (upper and lower) bounds on certain observables in the dual field theory. Such models have been used in the literature for modelling spontaneous and explicit breaking of translation invariance in holographic matter.

Identifying and understanding universal bounds on observables in Nature is certainly of great importance. Here, particular emphasis is give to the conjectured KSS bound for the shear viscosity to entropy density ratio, the lower bound on diffusion constants introduced by Hartnoll and extended by Blake, the upper bound on diffusion constants by Hartman et al as well as more recent bounds related to the speed of sound by Hohler et al. The authors carry out a set of numerical computations in order to test these bounds in field theories with (explicit or spontaneous) breaking of translations. These tests are new and, to the best of my understanding, correct. No new bounds have been introduced in this work.

At several points in the paper the authors claim that they have numerical ``proved" the conjectured bounds. The authors are reminded that numerically one can only disprove a conjecture by finding a counter-example. If the bound is not violated numerically, it simply means that in the specific model studied and for the specific values used, e.g. ($N,\mu, T,\dots$), the bound was respected. Having this in mind, the claims of the authors are too strong and should thus be toned down, in order to actually reflect the computations done. As the paper stands, the claims are too strong.

Let me now turn to more specific points (following the order of the paper): 1. It is well known that finite-N effects violate the KSS bound (but not finite $\lambda$ corrections). This itself suggests that this bound can not be a universal, regardless of the status of translation invariance. See 1108.0677 for a review. 2. $\tau_{eq}$ is the \textit{local} equilibration time and it's related to the applicability of hydrodynamics. This is different from the time needed to reach \textit{global} equilibrium, so please adjust the text. Furthermore, by now it's understood that the scale that sets $\tau_{eq}$ is not the distance to the first gapped quasinormal mode, but instead the location of the critical point (see e.g. 1904.12862) which can be much larger. How does this observation affect your results related to the upper bound? 3. For a system with both energy and charge density one needs to be more careful when defining the speed of sound, see e.g. 1205.5040 4. In the last sentence of Section 1, you say you will discuss the implications of the absence of a UV cutoff, but I can not find this discussion. Please indicate where this is done. 5. Please add a small appendix with the details of the numerical calculations that you carried out. This will add clarity and completeness to the paper and allow readers to follow easier. Also, add a few comments in the main text where you discuss the difference between explicit and spontaneous breaking in connection with the value of $N$ -- currently this is only done for the explicit case. 6. Figure 2, right panel: remove the y-axis label (since you display 2 different quantities on the same plot) 7. Page 7, point V, first 2 sentences: Essentially all the points on page 6 and 7 indicate that it's not clear what is the right definition for the shear viscosity when translations are broken. Given that, any conclusion for the violation or not of the KSS bound in this context can be attributed to simply using the wrong definition and thus not to carry any physical importance. This comment also applies to the orange line in Figure 2. I understand and I agree with the rest of the argument V regarding the mass of the shear mode. 8. Page 21: the fact that the stiffness becomes negative signals a dynamic instability. 9. It would be nice to add a table in your discussion section where you summaries the bounds and whether or not they are violated for spontaneous or explicit breaking of translations in your model. 10. The Hawking-Page transition has been associated to the confinement transition by Witten more than 10 years ago. How does this fit with the concept of the melting temperature? 11. Please check the text again for typographic errors. Furthermore, in several points the authors summarise the lack of deeper understanding of certain concepts in the form of questions, one after the other. This clearly shows their excitement for the topic, which is highly appreciated and valued, but it can be disruptive to the reader -- please consider formulating in a different way.

Given the above, I recommend this paper for publication in Sci.Post, after the authors have addressed the points raised here.

  • validity: good
  • significance: ok
  • originality: good
  • clarity: good
  • formatting: good
  • grammar: good

Author:  Matteo Baggioli  on 2020-06-21  [id 860]

(in reply to Report 1 on 2020-06-03)

We would like to thank both the referees for their detailed reports which led to several improvements in our manuscript. In the attached file we address one by one all the points raised.

We believe that our manuscript has improved a lot and we hope the referee will find it now suitable for publication in SciPost.

Attachment:

reply1.pdf

---

## Round 1 · Referee Report · Anonymous (Referee 2) · 2020-6-18

Report

This paper studies various bounds on transport proposed in the literature, with the hope of reaching something of universal status, all of which ultimately fall short depending on the context. Nevertheless I think such bounds may be interesting if the physical conditions under which they hold can be delineated.

The authors provide a fairly comprehensive review of such bounds. In addition they add new analyses in various cases: lower- and upper-bounds on diffusion, and bounds on the speed of sound. These computations apply to a particular class of holographic models with broken translations.

I think the paper is of generally high quality and the results are an interesting and original contribution to the area. I recommend that it be published, subject to the following minor changes:

1- "we have proved numerically that all the diffusive processes in the holographic models with broken translations considered obey the upper bound proposed in [15]". I think this presentation of their results is not appropriate - short of a proof they have simply not found any violation in their numerical studies.

2- "This confirms the idea that $\eta/s$ is not a meaningful quantity in non-relativistic systems" — The authors could clarify this and related statements in the manuscript. Do they relate only to translation-breaking as studied here, or to all non-relativistic contexts? e.g. what about Galilean-invariant systems?

3- There are several equations in the draft (8),(22),(33),(38) where for clarity the authors should indicate whether these are approximate or exact expressions in $k$. At the moment they are presented as exact.

4- The paper should include some technical details surrounding the holographic diffusion constant calculation. How is $D$ computed numerically? Is it a direct computation or is there some fitting taking place with (22)?

  • validity: high
  • significance: good
  • originality: good
  • clarity: good
  • formatting: excellent
  • grammar: good

Author:  Matteo Baggioli  on 2020-06-21  [id 861]

(in reply to Report 2 on 2020-06-18)

We would like to thank both the referees for their detailed reports which led to several improvements in our manuscript. In the following we address all the points.

To simplify their work of review, we have kept all the modifications in the manuscript in red color.

We believe that our manuscript has improved a lot and we hope the referee will find it now suitable for publication in SciPost.

Attachment:

reply2.pdf

---

## Round 5 · Author Response

We would like to thank the referees for their detailed report. We have carefully addressed all the points of both the referees.
We believe our manuscript has consistently improved.

---

## Round 5 · List of Changes

All the changes are left in red in the manuscript.

---

## Editorial Decision

published